# Shearing in flow environment promotes evolution of social behavior in microbial populations

Gurdip Uppal*, Dervis Can Vural*

Department of Physics, University of Notre Dame, Notre Dame, United States

**Abstract** How producers of public goods persist in microbial communities is a major question in evolutionary biology. Cooperation is evolutionarily unstable, since cheating strains can reproduce quicker and take over. Spatial structure has been shown to be a robust mechanism for the evolution of cooperation. Here we study how spatial assortment might emerge from native dynamics and show that fluid flow shear promotes cooperative behavior. Social structures arise naturally from our advection-diffusion-reaction model as self-reproducing Turing patterns. We computationally study the effects of fluid advection on these patterns as a mechanism to enable or enhance social behavior. Our central finding is that flow shear enables and promotes social behavior in microbes by increasing the group fragmentation rate and thereby limiting the spread of cheating strains. Regions of the flow domain with higher shear admit high cooperativity and large population density, whereas low shear regions are devoid of life due to opportunistic mutations.
DOI: https://doi.org/10.7554/eLife.34862.001

## Introduction

Cooperation is the cement of biological complexity. A combined investment brings larger returns. However, while cooperating populations are fitter, individuals have evolutionary incentive to cheat by taking advantage of available public goods without contributing their own. Avoiding the cost of these goods allow larger reproduction rates, causing cheaters to proliferate until the lack of public goods compromise the fitness of the entire population. In other words, while cooperating populations are fitter than non–cooperating ones, cooperation is not evolutionarily stable. How then can social behavior emerge and persist in microbial colonies?

The evolution of cooperation is an active field of research, with multiple theories resolving this dilemma (*Axelrod and Hamilton, 1981*; *Sachs et al., 2004*; *Sachs and Simms, 2006*; *Nowak, 2006*). According to (*Fletcher and Doebeli, 2009*) the fundamental mechanism is assortment. That is, in order for cooperation to evolve, cooperators and cheaters must experience different interaction environments.

How this assortment is achieved is a central question. Possibilities include positive and negative reciprocity (*Trivers, 1971*; *Clutton-Brock and Parker, 1995*; *El Mouden et al., 2010*), where cooperators are rewarded later by others, or where cheaters are inflicted a cost, via policing or reputation. For example, quorum signals reveal whether available public goods add up to the population density. In this case, altruists cut back public good production to eliminate cheaters (albeit with collateral damage) (*Allen et al., 2016*; *Sandoz et al., 2007*; *Diggle et al., 2007*). Another idea is group selection (*Wynne-Edwards, 1962*; *Haldane, 1932*; *Traulsen and Nowak, 2006*; *Wilson, 1975*) and its modern incarnation, multi-level selection, (*Wilson and Sober, 1994*) which propose that cooperating groups (or groups of groups) will reproduce faster than non-cooperating ones and prevail. Kin-selection theory (*Hamilton, 1964a*, *1964b*; *Williams, 1966*; *Smith, 1964*; *Hamilton, 1975*; *Lion et al., 2011*) provides a mechanism that arises from individual level dynamics. Kin-selection

*For correspondence:
guppal@nd.edu (GU);
dvural@nd.edu (DCV)

Competing interests: The authors declare that no competing interests exist.

**eLife digest** According to the principle of the 'survival of the fittest', selfish individuals should be better off compared to peers that cooperate with each other. Indeed, even though a population of organisms benefits from working together, selfish members can exploit the cooperative behavior of others without doing their part. These 'cheaters' then use their advantage to reproduce faster and take over the population.

Yet, social cooperation is widespread in the natural world, and occurs in creatures as diverse as bacteria and whales. How can it arise and persist then? One idea is that when individuals form distinct groups, the ones with cheaters will perish. Even though a selfish individual will fare better than the rest of its team, overall, cooperating groups will survive more and reproduce faster; ultimately, they will be favored by evolution. This is called group selection.

Here, Uppal and Vural examine how the physical properties of the environment can influence the evolution of social interactions between bacteria. To this end, mathematical models are used to simulate how bacteria grow, evolve and drift in a flowing fluid. These are based on equations worked out from the behavior of real-life populations.

The results show that flow patterns in a fluid habitat govern the social behavior of bacteria. When different regions of the fluid are moving at different speeds, 'shear forces' are created that cause bacterial colonies to distort and occasionally break apart to form two groups. As such, cooperative groups will rapidly form new cooperating colonies, whereas groups with cheaters will reproduce slower or perish.

Furthermore, results show that when different areas of the fluid have different shear forces, social cooperation will only prevail in certain places. This makes it possible to use flow patterns to fine tune social evolution so that cooperating bacteria will be confined in a certain region. Outside of this area, these bacteria would be taken over by cheaters and go extinct.

Bacteria are both useful and dangerous to humans: for example, certain species can break down pollutants in the water, when others cause deadly infections. These results show it could be possible to control the activity of these microorganisms to our advantage by changing the flow of the fluids in which they live. More broadly, the simulations developed by Uppal and Vural can be applied to a variety of ecosystems where microscopic organisms inhabit fluids, such as plankton flowing in oceanic currents.

DOI: https://doi.org/10.7554/eLife.34862.002

proposes that individuals cooperate with those to which they are genetically related, and thus, a cooperative genotype is really cooperating with itself.

Hamilton conjectured that kin selection should promote cooperation if the population is viscous, that is when the mobility of the population is limited (*Hamilton, 1964a*; *Hamilton, 1964b*). This helps ensure that genetically related individuals cooperate with each other. However, competition within kin can inhibit altruism (*Taylor, 1992*; *Wilson et al., 1992*). One solution to this is if individuals disperse as groups, also known as budding dispersal. This was shown to promote cooperation theoretically by (*Gardner and West (2006)* and demonstrated experimentally by (*Kümmerli et al., 2009*). Budding dispersal has also been studied by (*Pollock, 1983*; *Goodnight, 1992*; *Kelly, 1994*) and by (*Wilson et al., 1992*) from a group selection perspective.

There may be multiple and overlapping mechanisms underlying assortment. There is much debate in the literature over which theories best explain the evolution of cooperation and under which conditions each theory may be applicable. There is still not agreement, for example, on whether kin-selection and group selection can be viewed as equivalent theories (*Lion et al., 2011*; *Kramer and Meunier, 2016*). According to (*Simon et al., 2012*; *Simon et al., 2013*), since relatedness need not impact certain group level selection events, such as various games between groups, group selection is distinct from kin selection. Also, individual and group level selection events are generally asynchronous in nature and therefore cannot be equivalent. However, the debate still goes on (*Kramer and Meunier, 2016*; *West et al., 2007*; *Gardner, 2015*; *Goodnight, 2015*).

Typically, evolution of cooperation is quantitatively analyzed with the aid of game theoretic models applied to well-mixed populations, networks and other phenomenological spatial structures

(*Szabó and Fáth, 2007*; *Allen et al., 2013*; *Nowak and Sigmund, 2004*; *Vural et al., 2015*). While there are few models that take into account spatial proximity effects, (*Medvinsky et al., 2002*; *Nadell et al., 2010*; *Nadell et al., 2013*; *Dobay et al., 2014*; *Driscoll and Pepper, 2010*) and the influence of decay and diffusion of public goods (*Dobay et al., 2014*; *Wakano et al., 2009*; *Hauert et al., 2008*), how advective fluid flow influences social evolution remains mostly unexplored. The present study aims to fill this gap.

A flowing habitat can have a drastic effect on population dynamics (*Tél et al., 2005*; *Nickerson et al., 2004*; *Koshel' and Prants, 2006*; *Sandulescu et al., 2008*). For example, a flowing open system can allow the coexistence of species despite their differential fitness (*KarolyiKárolyi et al., 2000*). Interactions between fluid shear and bacterial motility has been shown to lead to shear trapping (*Rusconi et al., 2014*; *Berke et al., 2008*) which causes preferential attachment to surfaces (*Berke et al., 2008*; *Li et al., 2011*). Turbulent flows can also lead to a trade-off in nutrient uptake and the cost of locomotion due to chemotaxis (*Taylor and Stocker, 2012*), and can drastically effect the population density (*Pigolotti et al., 2012*; *Perlekar et al., 2010*). Most importantly, the reproductive successes of species (and individuals within a single species) are coupled over distance, through the secretion of toxins, goods, and signals (*Mimura et al., 2000*; *Allison, 2005*; *Hibbing et al., 2010*). The spatial distributions of all such fitness altering intermediaries depend on flow. Indeed, the experimental study by *Drescher et al. (2014)* has shown that flow can help promote cooperation in bacterial biofilms. Thus, we are motivated to find out how flow plays a role in the evolution of cooperation.

Here we theoretically study how fluid dynamics molds the social behavior of a planktonic microbial population. Qualitatively stated, our evolutionary model has three assumptions: (1) Individuals secrete one waste compound and one public good. The former has no cost, whereas the latter does. (2) Mutations can vary the public good secretion rates of microbes, thereby producing a continuum of social behavior. (3) Microbes and their secretions diffuse and flow according to the laws of fluid dynamics.

Under these assumptions, we find, through computer simulations and analytical theory, that bacteria self organize and form patterns of spots, which then exhibit an interesting form of budding dispersal when sheared by ambient fluid flow. The dispersal process preserves the group structure, thereby enabling evolutionarily stable social behavior.

Our model is applicable to a wide variety of social ecosystems ranging from phytoplankton flowing in oceanic currents to opportunistic bacteria colonizing blood or industrial pipelines. Our findings imply that greater social complexity amongst planktonic species would be observed in regions of large shear, such as by rocks and river banks. We might even speculate that multicellularity may have originated near fluid domains with large shear flow, rather than the bulk of oceans or lakes.

This paper is organized as follows: We first establish that under certain conditions our physical model gives rise to spatially organized cooperative structures. The structures are a natural byproduct of the dynamics of the system. Furthermore, these social structures reproduce in whole to form new identical structures. Variants of such structures have already been studied in ecological settings (*Tian et al., 2011*; *Camara, 2011*; *Baurmann et al., 2007*; *Wilson et al., 2003*) and growth patterns of microbial populations havebeen explored (*Ben-Jacob et al., 1994*; *Chang-Li et al., 1988*). We then study the effects of mutation. We first start with a simplified model with only two phenotypes: cheater and altruist. We then generalize to a continuum of public good secretion rates. In both cases, we observe that above a certain mutation rate, cheating strains take over groups which leads to total extinction. The latter finding is consistent with other empirical (*Rainey and Rainey, 2003*; *Diggle et al., 2007*) and theoretical studies (*Nowak and May, 1992*).Through the fragmentation of social groups, and death of cheating groups, we recover the results of Simpson's paradox (*Chuang et al., 2009*) where individual groups may decrease in sociality, but the population as a whole becomes more social.

After setting up the stage for naturally forming social groups, we present our central and novel finding, that flow shear can lead to evolutionarily stable cooperative behavior within the population. Specifically, we demonstrate and study the evolution ofsociality of a microbial population (1) subjected to constant shear, (2) embedded in a cylindrical laminar flow and (3) in a Rankine vortex. We find that in all three cases population density and cooperative behavior scales with flow shear.

The mechanism of action is that shear distortion enhances the fragmentation of cooperative clusters, thereby increasing the group fragmentation rate and limiting the spread of cheaters. If the

shear is large enough that groups are torn apart at a larger rate than the mutation rate, then cooperation will prevail. Otherwise, groups will become dominated by cheaters, and eventually die out (Figure 2, *Videos 1–3*).

## Results

We study a physically realistic spatial model of microorganisms, where fluid dynamical forces contribute significantly to the evolution of their social behavior. Our analysis consists of simulations and analytical formulas.

We simulate microbes as discrete particles subject to stochastic physical and evolutionary forces, and the compounds secreted by microbes as continuous fields. In contrast, our analytical expressions are derived from analogous equations that are entirely continuous and deterministic. In general, we should not expect the discrete simulations to perfectly be described by the continuous set of partial differential equations. Nevertheless, the continuous system of equations do allow us to obtain relevant quantities such as group size and group fragmentation rate to a good approximation (cf. appendix 1 and Figure 3).

In our model, the microorganisms secrete two types of diffusive molecules that influence each other's fitness (*Figure 1*). The first molecule, the concentration of which is denoted by $c_1(x, t)$, is a beneficial public good that increases the fitness of those exposed, whereas the second, $c_2(x, t)$, is a waste compound or toxin that has the opposite effect, and effectively acts as a volumetric carrying capacity. The continuous equations that represent our system are

$$\frac{\partial n}{\partial t} = d_b \nabla^2 n - \mathbf{v} \cdot \nabla n + n \left[ \alpha_1 \frac{c_1}{c_1 + k_1} - \alpha_2 \frac{c_2}{c_2 + k_2} - \beta_1 s_1 \right] + \mu \frac{\delta^2}{\delta s_1^2} n, \tag{1}$$

$$\frac{\partial c_1}{\partial t} = d_1 \nabla^2 c_1 - \mathbf{v} \cdot \nabla c_1 + \int_0^\infty n s_1 \mathrm{d} s_1 - \lambda_1 c_1, \tag{2}$$

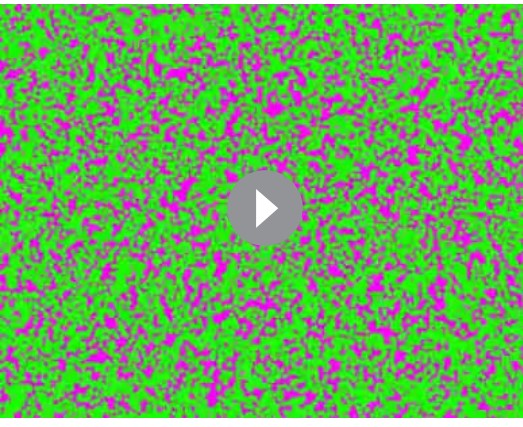

**Video 1.** This is a video file of a simulation of the homogeneous phase.
DOI: https://doi.org/10.7554/eLife.34862.003

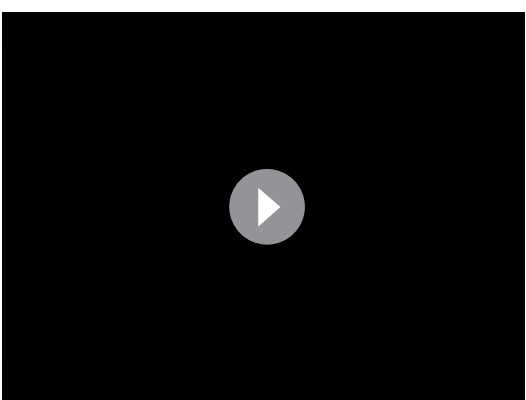

**Video 2.** This is a video file of a simulation of the group phase.
DOI: https://doi.org/10.7554/eLife.34862.004

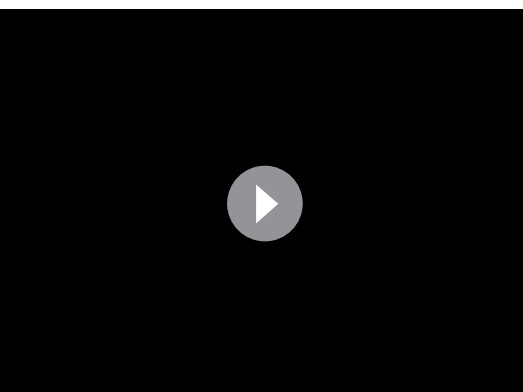

**Video 3.** This is a video file of a simulation of the group phase under Couette flow.
DOI: https://doi.org/10.7554/eLife.34862.005

$$\frac{\partial c_2}{\partial t} = d_2 \nabla^2 c_2 - \mathbf{v} \cdot \nabla c_2 + s_2 \int_0^\infty n \mathrm{d}s_1 - \lambda_2 c_2. \tag{3}$$

Here $n$ is a shorthand for $n(x, t, s_1)$, the number density of microbes at time $t$ and position $x$ that produce the public good at a rate of $s_1$. These microbes pay a fitness cost of $\beta_1 s_1$ per unit time. The production rate of waste $s_2$ on the other hand, is assumed constant for all, and has no fitness cost. Waste limits the number of individuals that a unit volume can carry. Microbes secreting public goods at a rate $s_1$ replicate to produce others with the same secretion rate. This reproduction rate is given by the square bracket. However, the production rate $s_1$ can change due to mutations. This is described by the last term of *Equation 1*. Mutations can be thought as diffusion in $s_1$ space.

In all three equations the first two terms describe diffusion and advection, while the last two terms of *Equation 2* and *Equation 3* describe the production and decay of chemicals. The first two terms in the square bracket describe the effect of the secreted compounds on fitness. This saturating form is experimentally established and well understood (*Monod, 1949*). The crucial third term in the square bracket describes the cost of producing the public good, which increases linearly.

## Social groups as turing patterns

Diffusion can cause an instability that leads to the formation of intriguing patterns (*Turing, 1990*), which among other fields, have been investigated in ecological context (*Tian et al., 2011*; *Camara, 2011*; *Baurmann et al., 2007*; *Wilson et al., 2003*). These so called Turing patterns typically form when an inhibiting agent has a diffusion length greater than that of an activating agent. For our model system, the waste compound and public good play the role of inhibiting and activating agents, and patterns manifest as cooperating microbial clusters *Figure 2*. The size and reproduction rate of these clusters, in terms of system parameters, can be estimated from a Turing analysis (cf. appendix 1). *Figure 3* shows the values of diffusion constants that gives rise to Turing patterns, as well as the size of the groups.

In the homogeneous phase, the system is evolutionarily very unstable, since as soon as one cheating mutant emerges, it quickly takes over the entire population, ultimately causing the population to go extinct (*Video 1*). The group phase tolerates cheaters better, since once a cheater emerges it will take over and compromise the fitness of only one group, while the others will live on. However, in the absence of group fragmentation, novel cheating mutations will ultimately emerge in all groups, and annihilate the population one group at a time (*Video 2*).

The main contribution of this paper is to demonstrate that stable social cooperation can be induced or enhanced by fluid flow gradients. Specifically, shear forces induced by advective flow distorts and fragments microbial clusters, leading to a kind of budding dispersal, which in turn enables evolutionary stable cooperation (*Figure 2*, *Video 3*).

It is well known that evolutionary outcomes can depend on individuals being discrete (*Durrett and Levin, 1994*). In our model, having a continuous population density can allow for the existence of 'micro-mutant populations' which can spread easier between adjacent groups. The discreteness further separates the clusters of microbes from each other, since there cannot exist fractional individuals. In reality microbes are quantized, and we thus expect a discrete simulation to better model the biology. In *Figure 3* we present the phase diagram of the system, as obtained by analytical theory, discrete agent based simulations (where microbes are discrete, self-replicating brownian particles), and continuous simulations (where *Equations 1,2,3* are solved numerically). In order not to obfuscate the biology, we report our detailed mathematical treatment in the appendix.

## Effect of shear on groups

We quantitatively determine the effect of different flow velocity profiles on the social evolution of the system. A constant fluid flow merely amounts to a change in reference frame, which of course, does not change the evolutionary fate of the population. However, we find that velocity gradients cause significant changes to the social structure, both spatially and temporally. Specifically, we find that large shear rate causes microbial groups to distort and fragment, which in turn facilitates group reproduction. To investigate this effect in detail, we ran simulations for three fluid velocity distributions: Couette flow, Hagen-Poiseuille flow, and Rankine vortex.

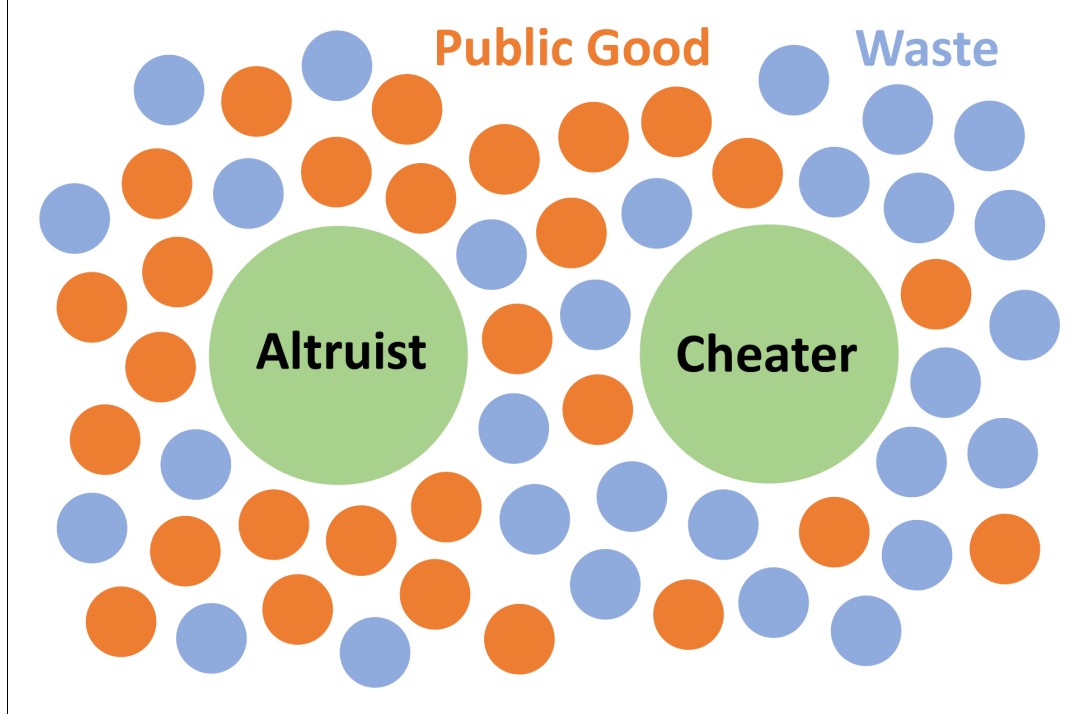

**Figure 1.** Schematics of our Model. The microbes secrete two types of molecules into the environment. The first, a beneficial public good that promotes growth, and the second, a waste or harmful substance hinders growth. Cheating microbes produce lessor none of the former, while benefiting from public goods secreted by the altruistic population.

DOI: https://doi.org/10.7554/eLife.34862.006

## Evolution of sociality in constant shear for a binary phenotype

We first look at a simplified system with just two phenotypes, cheater and altruist, to gain a basic understanding of the mechanism involved. Mutations can cause a switch in social behavior.

To see the effect of shear on social evolution, we introduced Couette flow to the microbial habitat. In this case, the flow velocity takes the form

$$\mathbf{v}(\mathbf{x}) = v_{\max} \frac{r}{R} \hat{z},$$

where $R$ is the radius of the pipe, and $z$ is the longitudinal direction.

The shear rate is the derivative of the flow velocity and is related to the maximum flow rate $v_{\max}$,

$$\sigma = \frac{dv}{dr} = \frac{v_{\max}}{R}.$$

We ran simulations for various shear rates and diffusion constants and observed that shear does not significantly influence the *region* of parameter space that gives rise to cooperating groups. However *if* the system parameters are conducive to the formation of groups, shear tears groups apart and *increases* the rate at which spatially distinct cooperative clusters form.

We find that the group fragmentation rate $\omega(\sigma)$, depends linearly on the shear rate $\sigma$ (*Figure 4*)

$$\omega(\sigma) = m\sigma + \omega_0,$$

where $\omega_0$ is the fragmentation rate solely due to microbial diffusion and can approximately be given by the Turing eigenvalue $\omega_0 \approx \Lambda_{\max}$ (see appendix). The constant of proportionality $m$ is given empirically from our simulations and depends on diffusion lengths and group density. This holds in the low density regime. Once the population density becomes large, group-group interactions slow the group reproduction rate and the population saturates.

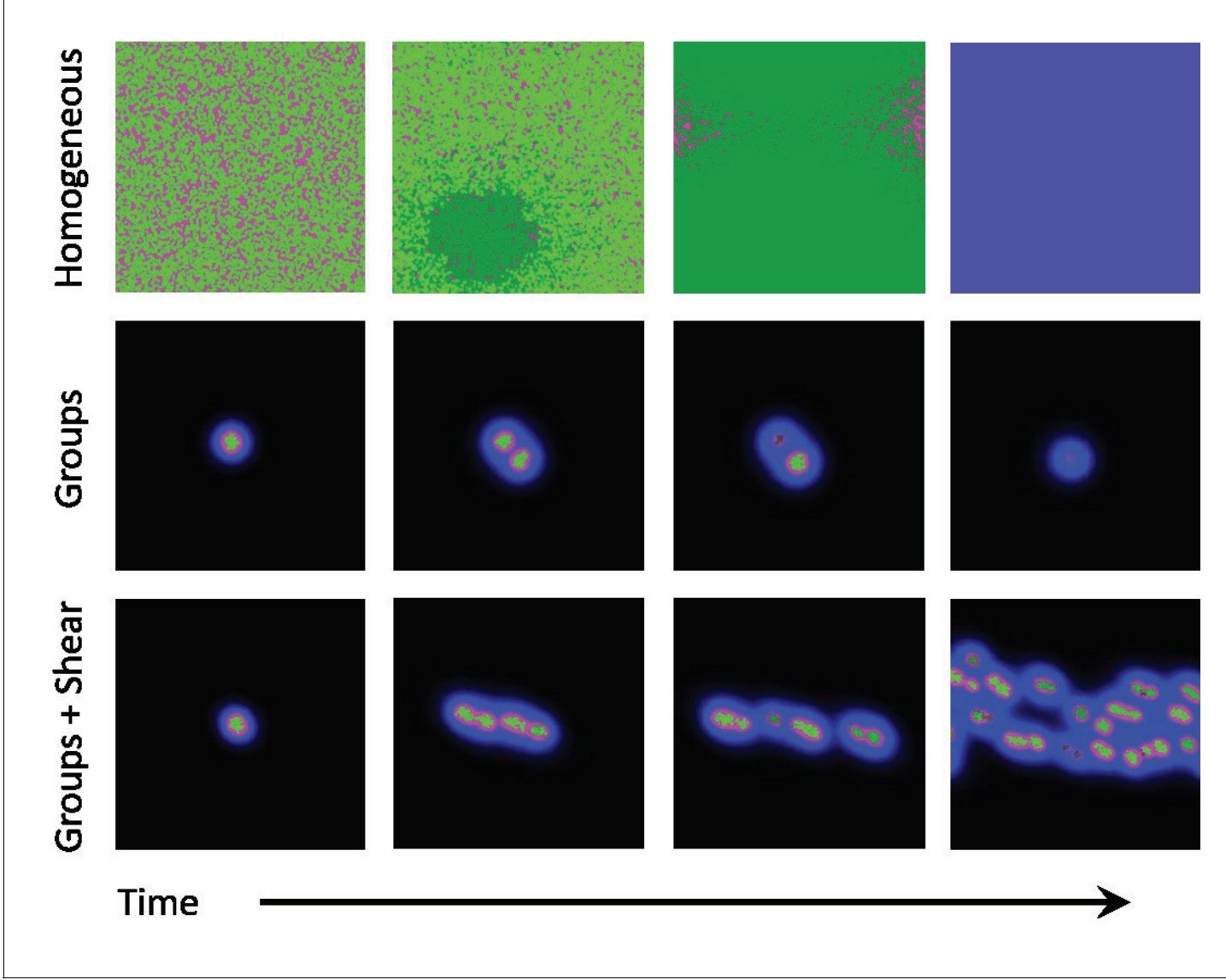

**Figure 2.** Snapshots of homogeneous and group phases without and with shear. Microbes interact by secreting diffusive chemicals into their environment. Cooperators are seen as bright green dots, and cheaters are seen as dark green dots. The waste compound is shown as blue and the public good as red, the two combined is seen as magenta. Top row: In the homogeneous phase the microbes spread to fill the domain. Cheaters quickly begin to take over, and eventually take over the whole domain. With no cooperators left, the public good decays away and the system goes extinct. Middle row: In the group phase, when the diffusion length of the waste compound is larger than the diffusion length of public good, microbes form stable groups. As the microbes increase in number, the groups split apart and form new groups. As mutations occur within groups, the cheaters take over and the group goes extinct. Cooperation can only be stable here if groups reproduce quicker than mutants take over. Bottom row: By adding a shearing flow to the group system, we can cause the groups to split apart quicker. Mutations still take over groups, but the groups are able to reproduce quicker than mutants take over, thus allowing cooperative groups to prevail at steady state. The simulation videos corresponding to this figure are provided in the *Videos 1–3*.
DOI: https://doi.org/10.7554/eLife.34862.007

Larger shear corresponds to a faster rate of group fragmentation, thus enabling or enhancing social behavior in the microbial population. Given the group size and fragmentation rate as a function of shear, we can calculate roughly where the critical shearrate is for sociality. We also find that the group population $N$ increases linearly with shear,

$$N(\sigma) = n\sigma + N_0.$$

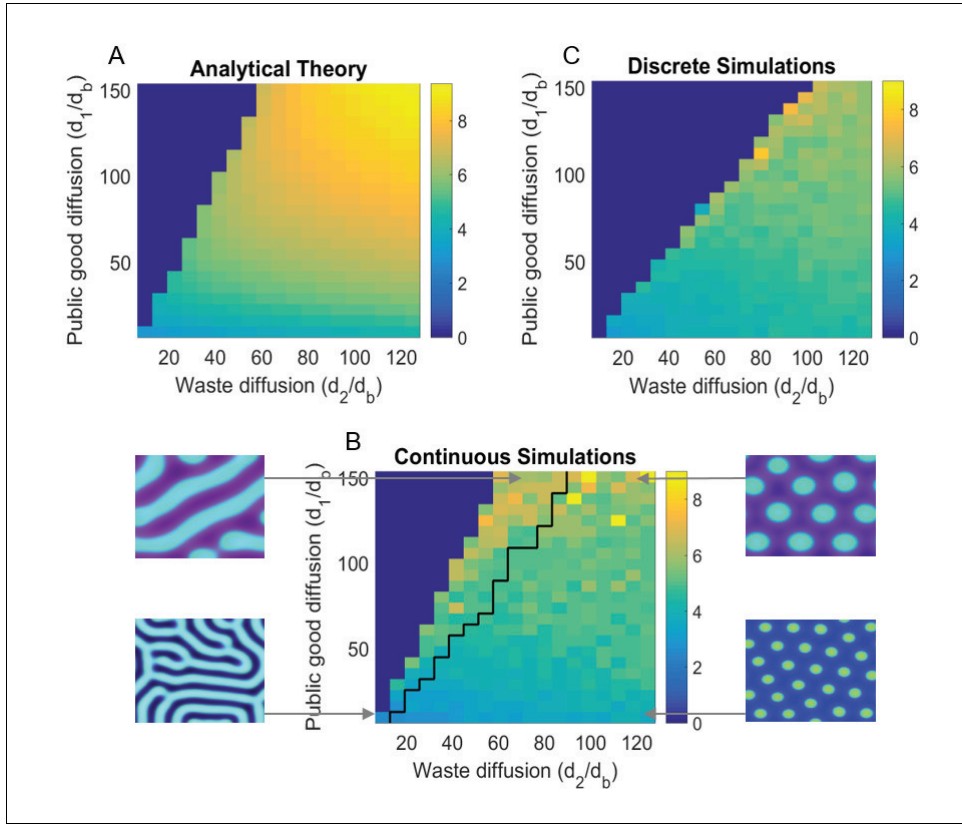

**Figure 3.** Turing analysis results. The top-left figure, (**A**) shows the group size $2\pi/k_{\text{fast}}$ as obtained by our theoretical analysis (appendix); whereas the bottom figure, (**B**) shows the same for continuous simulations, and the top-right figure, (**C**) is for agent based simulations. The black line in (**B**) divides parameters that give rise to striped patterns, and those that give rise to spots, corresponding images are shown to the left and right of (**B**). Due to the discreteness of the agent based simulations, Turing patterns are not always stable where they might be in the continuous analogue. We see that the discrete simulations cut off around where we would see stripes in the continuous case, and do not see striped patterns in the discrete case. For different sets of parameters, we can also see Turing patterns in the discrete-stochastic case where they might not occur in the continuous case. For the region where Turing patterns are stable, the continuous theory gives a good prediction of group size and group reproduction rate. The Matlab code and data for this figure is provided in *Figure 3—source data 1*.

DOI: https://doi.org/10.7554/eLife.34862.008

The following source data is available for figure 3:

**Source data 1.** Matlab data and code files for *Figure 3*.
DOI: https://doi.org/10.7554/eLife.34862.009

where $n$ and $N_0$ are the slope and intercept of the line in *Figure 4B*. This also influences where the critical shear rate will be.

Cooperation is stable if a group is able to fragment before a cheating strain emerges and proliferates in the group. Therefore, for stability, we need the take-over time to exceed the time it takes for a group to reproduce. A mutant emerges at a rate of $\mu N(\sigma)$ (we emphasize that $\mu$ is not the generic mutation rate, but the rate at which a particular social gene mutates). Once a mutant emerges, it takes some time $\tau_d$ to spread to where the daughter group forms. $\tau_d$ will depend on where the mutant first emerges. Assuming a uniform distribution, and taking the diffusion time in two dimensions as a function of radius $r$, $\tau_d(r) = r^2/4d_b$, we obtain

$$<\tau_d> = \int_0^R \frac{r^2}{4d_b}\frac{2r}{R^2}\,\mathrm{d}r = \frac{R^2}{8d_b},$$

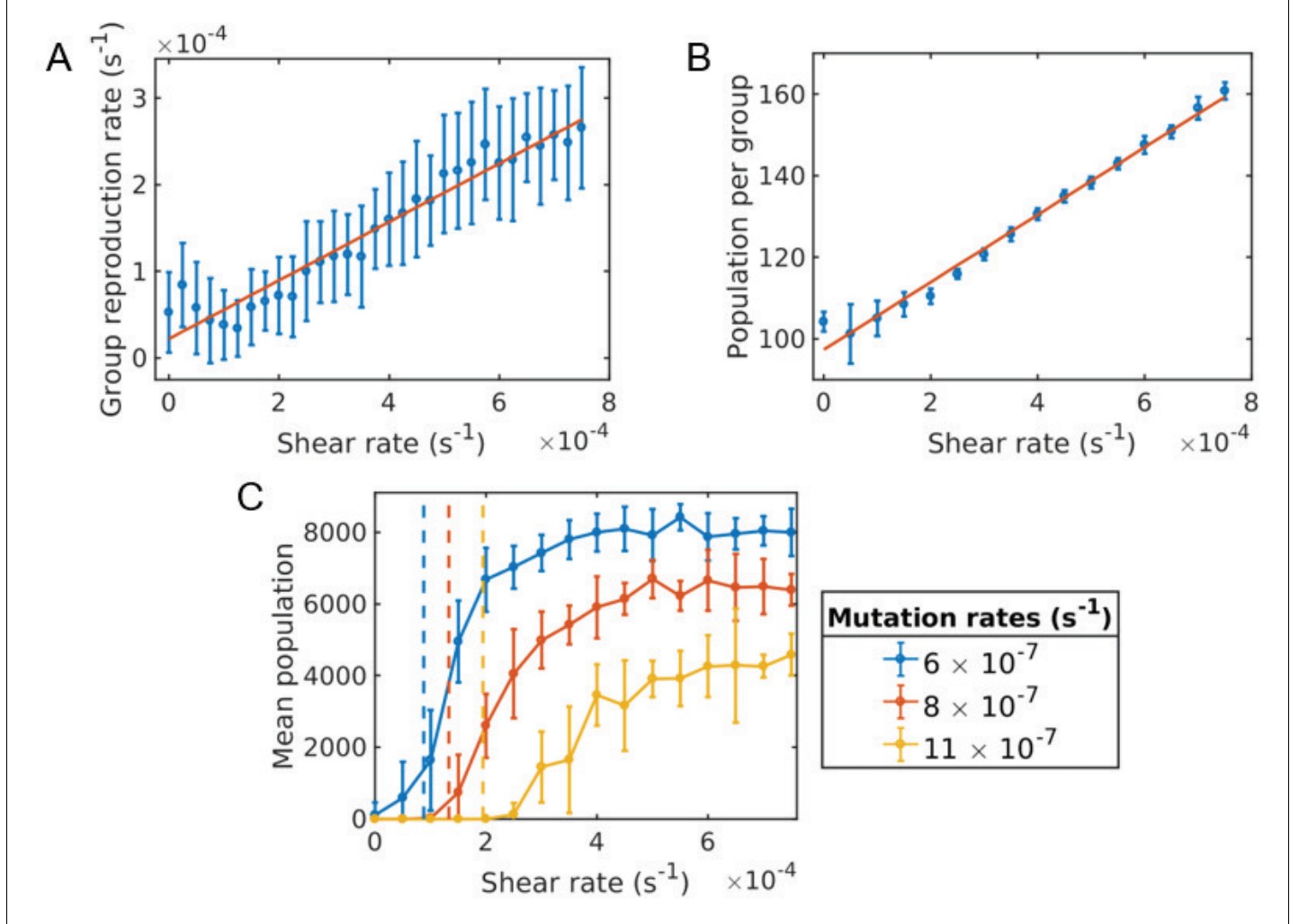

**Figure 4.** Critical shear for cooperativity for cheater-altruist system. (**A**) Group fragmentation rate versus shear rate. We see that the group fragmentation rate increases linearly with the shear rate. (**B**) Group population versus shear rate. As the shear distorts and elongates the group, the average group population also increases linearly with shear. (**C**) Average population versus shear rate for different mutation rates. Simulations were run for a time of $5.0 \times 10^5$ s and averaged over 10 runs for each shear rate and mutation rate. Error bars correspond to one standard deviation. Here, a mutation corresponds to a full cheater, with no public good secretion. The population goes extinct under larger mutation rates unless the shear rate is above the critical value. The critical shear values for different mutation rates are roughly obtained by 10 and are shown by the vertical dashed lines corresponding to curves of the same color. The Matlab code and data for this figure is provided in *Figure 4—source data 1*.

DOI: https://doi.org/10.7554/eLife.34862.010

The following source data is available for figure 4:

**Source data 1.** Matlab data and code files for *Figure 4*.

DOI: https://doi.org/10.7554/eLife.34862.011

where $R$ is the group radius. The take-over rate is then given by taking the inverse of the total take-over time, and the critical shear rate $\sigma_c$ necessary for social cooperation is given by equating the take-over rate with the reproduction rate,

$$\omega(\sigma_c) = \left[ \frac{1}{\mu N(\sigma_c)} + <\tau_d> \right]^{-1}. \tag{4}$$

The critical shear rate $\sigma_c$ above which the system can maintain stable cooperation is then given by the positive root,

$$\sigma_c = \frac{-b_\sigma + \sqrt{b_\sigma^2 - 4a_\sigma c_\sigma}}{2a_\sigma} \tag{5}$$

where $a_\sigma = <\tau_d>\mu mn$, $b_\sigma = <\tau_d>\mu n\omega_0 + <\tau_d>\mu mN_0 + m - \mu n$, and $c_\sigma = <\tau_d>\mu N_0\omega_0 + \omega_0 - \mu N_0$.

The values obtained from *Equation 5* is indicated by the vertical dashed lines in *Figure 4* and agrees with the computationally observed critical shear reasonably well. It may be possible to improve this formula further by taking into account additional factors, such as the non-uniform spatial distribution of population within a group and the elongation of groups with shear. Furthermore, as the mutants increase in numbers it becomes more likely that one of them crosses over the daughter group, thereby reducing further the expected $<\tau_d>$. We see better agreement with analytical theory and simulations at lower mutation rates, since these corrections are mainly to the diffusion time $<\tau_d>$, and become more significant at higher mutation rates, where $<\tau_d> \gg 1/\mu N$, (cf. *Equation 4*).

If shear is below the critical value (*Equation 5*), the system will be in a non-social state. Ultimately, cheaters will take over, and wipe out all groups. When shear is increased above the critical value however, the system will transition to a stable social state, thereby maintaining its fitness and dense population indefinitely. *Figure 4* shows the long-time population of the system versus the shear rate. The population goes extinct under larger mutation rates unless the shear rate is above the critical value.

When is shear necessary, and when is it just a sufficient condition for cooperation? By setting $\sigma_c = 0$ in *Equation 5* and solving for $\mu$ we can also obtain the critical mutation rate above which shear is necessary in order to have social cooperation. We get $\mu_c = 6.9 \times 10^{-7}$ analytically and our simulations show a critical mutation rate around $\mu_c = 5.5 \times 10^{-7}$.

## Evolution of sociality in constant shear for a continuum of phenotypes

As we will see, we obtain similar results when the available phenotypes include a continuum of social behaviors. In this case, a mutation changes the secretion rate of a microbe by a uniformly chosen random number between 0 and 1 s$^{-1}$.

We observe from our simulations, that mutations that increase the secretion rate of a microbe do not fixate, since the microbe now pays a higher cost and is less fit than its neighbors. However, once a mutation that lowers the secretion rate of a microbe occurs within a group, it quickly takes over the entire group, leaving individual groups homogeneous in secretion rate.

We show the diversity of social behaviors across groups and within individual groups in greater detail in *Appendix 1—figure 1*. Since less cooperative phenotypes always dominate more cooperative phenotypes, we find no diversity of social behavior within a group. However we do see a large variation across groups, which increases with shear.

Groups with different secretion rates reproduce at different rates. Groups with too low of a secretion rate are not stable and die off. In general, the system will evolve to a distribution of groups with secretion rates centered around the value of secretion rate that maximizes the group reproduction. For larger mutation rates, the system will tend towards lower average secretion rate and/or go extinct. The average secretion rate of the population can be maintained at a higher value by introducing shear flow, *Figure 5*.

We therefore have the same qualitative result as in the two phenotype case, if shear is below some critical value, the system will be in a non-social state. Ultimately, cheaters will take over, and wipe out all groups. As before, the social state of the population can transition from a non-cooperative state to a cooperative one with increased flow shear.

## Evolution of sociality in a flowing pipe

We now further generalize our results by looking at laminar flow with fixed boundaries, and with a continuum of public good secretion rates. For Hagen-Poiseuille flow, the shear rate varies linearly with the radius, taking its maximum value adjacent to the boundaries, when $r = R$. The flow and shear profiles are given as,

$$v = v_{\max}\left(1 - \frac{r^2}{R^2}\right), \quad \frac{\mathrm{d}v}{\mathrm{d}r} = -2v_{\max}\frac{r}{R^2}.$$

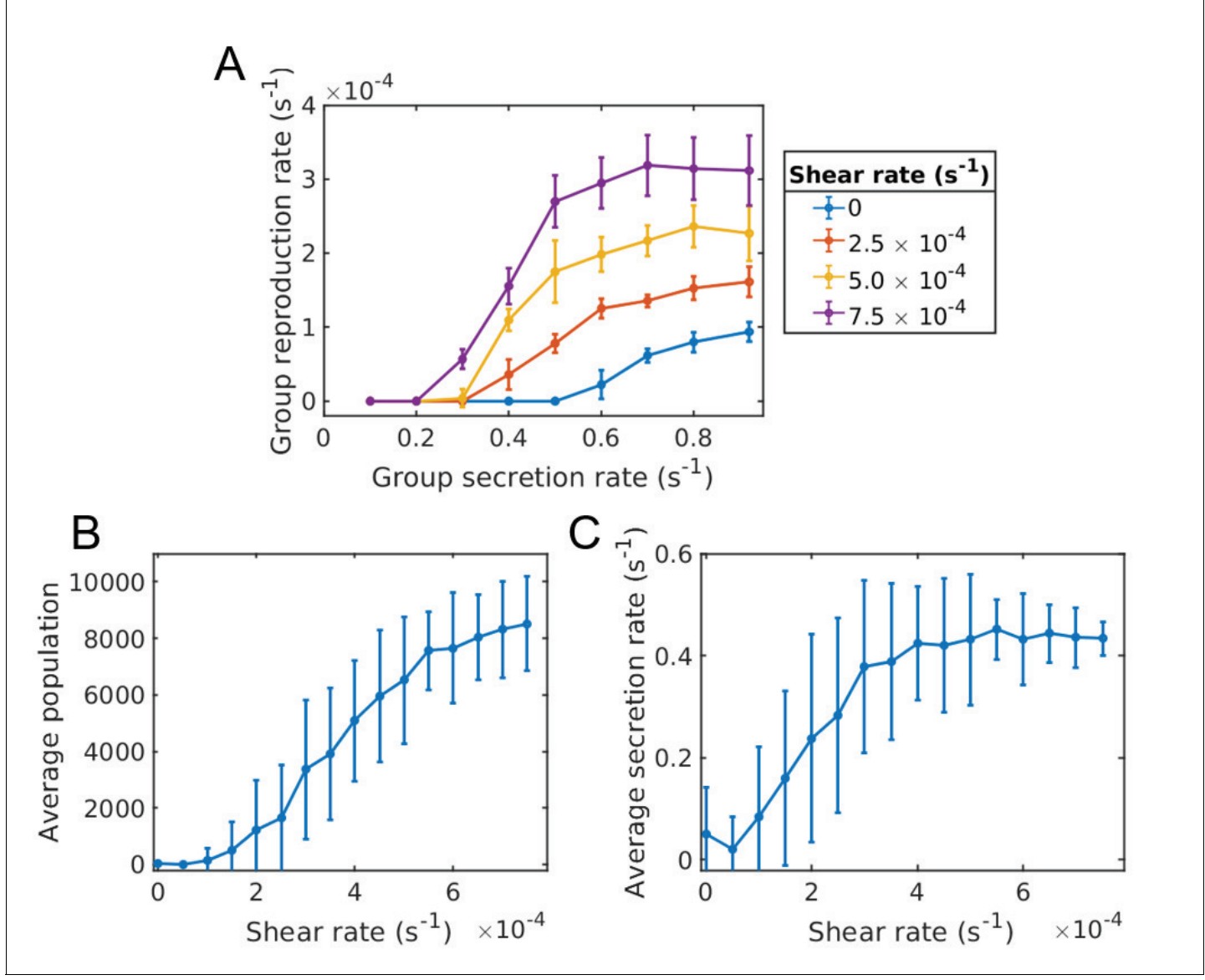

**Figure 5.** Evolution of sociality in constant shear (continuum of secretion rates). Individual groups are essentially homogeneous in secretion space, whereas the meta-population contains a distribution of groups with different secretion rates (*Appendix 1—figure 1*). (**A**) Groups that have a higher secretion rate reproduce quicker than those of lower secretion rate. Shear works to increase the reproduction rate of groups. (**B**) Just as in the two phenotype case, the population of the system increases with shear, since groups are able to split apart quicker than novel cheating mutations occur. (**C**) The average secretion rate of the entire population generally increases with shear and saturates around where the group reproduction rate is sufficient to maintain the population. Simulations were run for a time of $2.0 \times 10^5$ s and averaged over 80 runs for each shear rate and under a mutation rate of $\mu = 1.6 \times 10^{-6} \mathrm{s}^{-1}$. Error bars correspond to one standard deviation. The Matlab code and data for this figure is provided in *Figure 5—source data 1*.
DOI: https://doi.org/10.7554/eLife.34862.012

The following source data is available for figure 5:

**Source data 1.** Matlab data and code files for *Figure 5*.
DOI: https://doi.org/10.7554/eLife.34862.013

We therefore expect to see groups fragment quicker at the boundary, leading to larger cooperation, higher average secretion rate, and larger population, which is indeed what we do see (*Figure 6*, *Video 4*).

Earlier studies have proposed and shown that shear trapping due to the interaction between bacterial motility and fluid shear can result in preferential attachment to surfaces, (*Rusconi et al., 2014*;

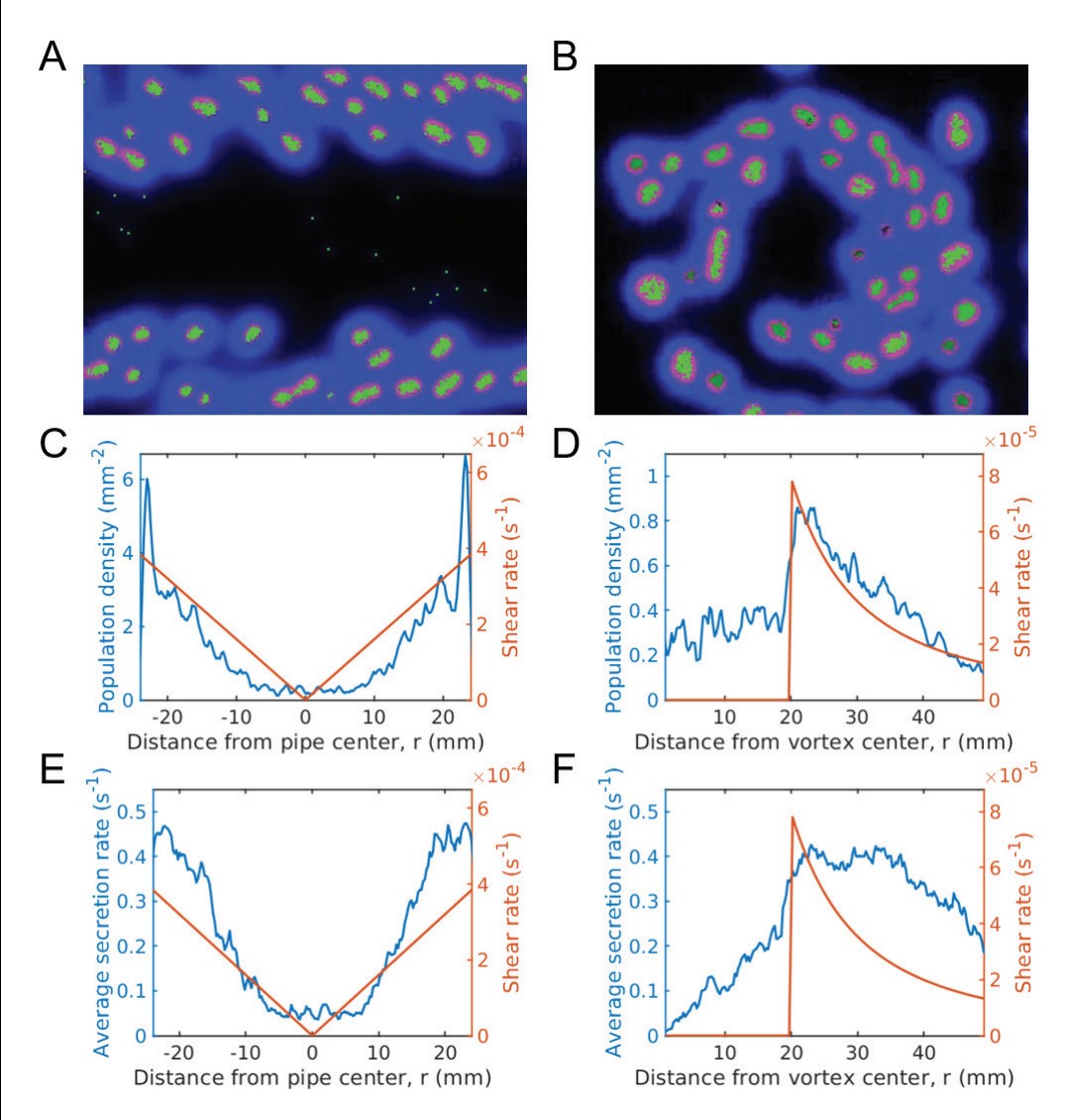

**Figure 6.** Evolution of sociality in pipe and vortex geometries (continuum of secretion rates). The top row gives simulation snapshots of the system in a Hagen-Poiseuille flow in a pipe (**A**) and of the system in a Rankine vortex (**B**). The middle row gives the average microbial population and the shear rate magnitude versus distance from the center of the pipe (**C**) and the center of the Rankine vortex (**D**). Since shear is spatially dependent, the population is localized in regions of large shear. For Hagen-Poiseuille flow, we see that the population is larger at the boundaries, where the shear is also larger (**C**). This is because groups fragment quicker at the boundaries and are able to overcome take-over by mutation, whereas near the center they cannot. For the Rankine vortex we also see that the population follows very closely to the shear (**D**), which suggests that the growth is proportional to shear. We caution that this holds in the low density limit. At higher densities the population saturates and is no longer proportional to shear. The bottom row gives the average public good secretion rates of the entire population for Hagen-Poiseuille flow (**E**) and for Rankine vortex flow (**F**). Again, regions of larger shear admit more cooperative populations with larger public good secretion rates. Simulations were run for a duration of $2.0 \times 10^5$ s under a mutation rate of $\mu = 2 \times 10^{-6} \mathrm{s}^{-1}$ and data was averaged over 200 runs. The undulations observed in the population plots are due to the finite size of the groups. Groups form layers of width equal to the group diameter. The population curve therefore shows undulations of width equal to the group width. Simulation videos of Hagen-Poiseuille flow and Rankine vortex flow are provided in *Videos 4, 5* and Matlab code and data for (**C**)-(**F**) is given in *Figure 6—source data 1*.

DOI: https://doi.org/10.7554/eLife.34862.014

The following source data is available for figure 6:

**Source data 1.** Matlab data and code files for *Figure 6*.

DOI: https://doi.org/10.7554/eLife.34862.015

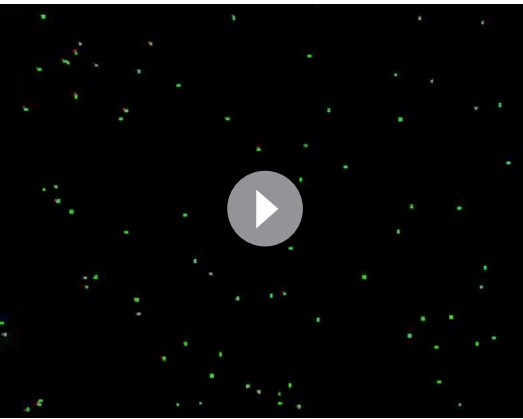

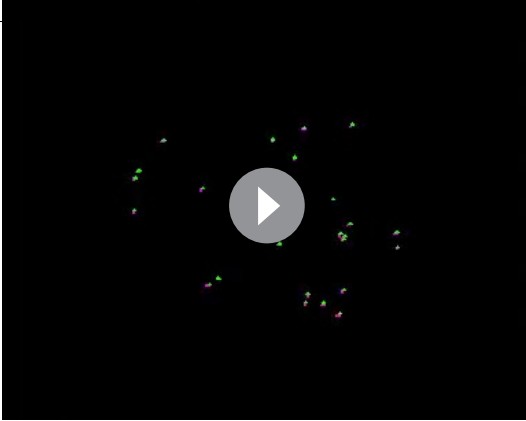

**Video 4.** This is a video file of a simulation of the group phase under a Hagen-Poiseuille flow.
DOI: https://doi.org/10.7554/eLife.34862.016

**Video 5.** This is a video file of a simulation of the group phase under a Rankine vortex flow.
DOI: https://doi.org/10.7554/eLife.34862.017

*Berke et al., 2008*; *Li et al., 2011*). In a similar spirit, we suggest that inhabiting surfaces may have the additional advantage of enhanced sociality, due to shear driven group fragmentation and dispersal.

## Evolution of sociality in vortices

In a vortex, the region above the critical shear value constitutes an annulus. Thus, we expect social behavior to be localized. Any point in the fluid outside this annulus will be taken over and destroyed by cheaters. In our simulations, at steady state we indeed see clusters whirling around exclusively within annulus, neither too near, nor too far from the vortex core (*Video 5*). Life cannot exist outside this annulus, as cheaters kill these groups.

The Rankine vortex in two dimensions is characterized by a vortex radius $R$ and a rotation rate $\Gamma$. The shear rate acting on a group acts tangential to the flow. The velocity profile and shear magnitude are given as,

$$\mathbf{v} = \begin{cases} \frac{\Gamma r}{2\pi R^2}\hat{\theta}, & r \leq R \\ \frac{\Gamma}{2\pi r}\hat{\theta}, & r > R \end{cases} \qquad \sigma = \begin{cases} 0, & r \leq R \\ \frac{\Gamma}{2\pi r^2}, & r > R \end{cases}$$

where $r^2 = x^2 + y^2$.

The shear rate is then a maximum at the minimum value of $r$ which occurs at the vortex radius $R$. We therefore expect to see the largest concentration of groups at the vortex radius, which is what we observe in our simulations (*Figure 6*).

## Limitations

While we paid close attention to physical realism, we also made important simplifying assumptions which under certain circumstances, may lead to incorrect conclusions. We caution the reader by enumerating the limitations of our model. First, since many microorganisms live in a low Reynolds number environment, we have chosen to neglect the inertia of microorganisms. However in reality, the microorganisms influence the flow around them. This effect will be particularly significant for a dense microbial population, especially when the microbes stick onto one other, or integrate via extracellular polymers. A more sophisticated model would include the coupling of the microbes to the flow. Secondly, the finite size and shapes of the microorganisms have been neglected. Instead, we have treated microbes as point particles, which will also invalidate our model in the dense population limit. Lastly, real microbes display a large number of complex behaviors such as biofilm formation and chemotactic migration. Here we have ignored the active response of microorganisms to the chemical gradients that surround them and to the surfaces they might attach and migrate. Instead, we took them as simple Brownian particles.

**Table 1.** Summary of system parameters.

| Parameter | Definition | Values |
|---|---|---|
| $d_b$ | Microbial diffusion constant | $0.3906 \times 10^{-6} \mathrm{cm^2 s^{-1}}$ |
| $d_1$ | Public good diffusion constant | $(1 \text{ to } 60) \times 10^{-6} \mathrm{cm^2 s^{-1}}$ |
| $d_2$ | Waste diffusion constant | $(1 \text{ to } 50) \times 10^{-6} \mathrm{cm^2 s^{-1}}$ |
| $v$ | Flow velocity | $(0 \text{ to } 100) \times 10^{-5} \mathrm{cm s^{-1}}$ |
| $\lambda_1$ | Public good decay constant | $5.0 \times 10^{-3} \mathrm{s^{-1}}$ |
| $\lambda_2$ | Waste decay constant | $1.5 \times 10^{-3} \mathrm{s^{-1}}$ |
| $k_1$ | Public good saturation | $0.01$ |
| $k_2$ | Waste saturation | $0.10$ |
| $s_1$ | Public good secretion rate | $(0 \text{ to } 1.0) \mathrm{s^{-1}}$ |
| $s_2$ | Waste secretion rate | $(0 \text{ to } 1.0) \mathrm{s^{-1}}$ |
| $\alpha_1$ | Benefit of public good | $7.5 \times 10^{-3} \mathrm{s^{-1}}$ |
| $\alpha_2$ | Harm of waste compound | $8.0 \times 10^{-3} \mathrm{s^{-1}}$ |
| $\beta_1$ | Cost of secretion | $0.2$ |
| $\mu$ | Mutation rate | $(6.0 \text{ to } 20.0) \times 10^{-7} \mathrm{s^{-1}}$ |

DOI: https://doi.org/10.7554/eLife.34862.018

## Discussion

It is well known that spatial structure is crucial in the evolution of cooperation, (*Wilson et al., 1992*; *Taylor, 1992*; *Lion and Baalen, 2008*). Many of these studies introduce these mechanisms 'manually', for example density regulation and migration are enforced by applying carrying capacities and migration rates to groups. In this study we have distanced ourselves from the typical game theoretic abstractions used to investigate evolution of cooperation. Instead we adopted a mechanical point of view. We investigated in detail, the fluid dynamical forces between microbes and their secretions, to understand how cooperation evolves among a population of planktonic microbes inhabiting in a flowing medium. In our first principles model, the spatial structuring and dispersion occur naturally from the physical dynamics.

We found that under certain conditions, microbes naturally form social communities, which then procreate new social communities of the same structure. More importantly, we discovered that regions of a fluid with large shear can enhance the formation of such social structures. The mechanism behind this effect is that fluid shear distorts and tears apart microbial clusters, thereby limiting the spread of cheating mutants. Our proposed mechanism can be seen as shear flow enhanced budding dispersal. This can also be viewed under the phenomenon of Simpson's paradox (*Chuang et al., 2009*) where individual groups may decrease in sociality, but the population as a whole becomes more social.

In our investigation, we found only certain regions of the fluid domain admits life, social or otherwise, as governed by the domain geometry and flow rate. From this perspective, it appears that evolution of sociality is a mechanical phenomenon.

In our physics-based model, groups emerge from individual-level dynamics and selection. Groups with cheaters are negatively selected, and give way to those without cheaters. On the other hand the ensemble of groups do not exhibit any variation in their propensity to progenerate cheaters, neither is such propensity heritable. Rather, the progeneration of cheating is a (non-genetic) *symptom* that inevitably manifests in every group that has been around long enough. In this sense, it might be appropriate to view the emergence and spread of cheaters in a microbial population as a phenomenon of 'aging', in the non-evolutionary and mechanical sense, that any system consisting of a large number of interdependent components will inevitably and with increasing likelihood, fall apart (*Vural et al., 2014*).

## Materials and methods

The analytical conclusions we derive from our system (*Equations 1,2,3*) has been guided and supplemented by an agent based stochastic simulation. Videos of simulations are provided in *Videos 1–5*. Our simulation algorithm is as follows: at each time interval, $\Delta t$, the microbes (1) diffuse by Brownian motion, with step size $\delta$ derived from the diffusion constant and a bias dependent on the flow velocity, $\delta = \sqrt{4d_b\Delta t} + v\Delta t$, (2) secrete chemicals locally that then diffuse and advect using a finite difference scheme, and (3) reproduce or die with a probability dependent on their local fitness given by $f = \Delta t\left[\alpha_1\frac{c_1}{c_1+k_1} - \alpha_2\frac{c_2}{c_2+k_2} - \beta_1 s_1\right]$. If $f$ is negative, the microbes die with probability 1, if $f$ is between 0 and 1 they reproduce with probability $f$, and if $f$ is larger than 1, they produce number of offspring given by the integer part of $f$ and another with probability given by the decimal part of $f$. Upon reproduction, random mutations may alter the secretion rate of the public good –and thus the reproduction rate– of the microbes. Mutations occur with probability $\mu$ and can change the secretion rate by a random number between 0 and $s_1$. The secretion rate is assumed to be heritable, and constant in time. Numerical simulations for figures were performed by implementing the model described above using the Matlab programming language and simulated using Matlab (Mathworks, Inc.). The source code for discrete simulations is provided in *Source code 1* and the source code for continuous simulations used in *Figure 3B* is provided in *Source code 2*.

A summary of the system parameters is given in *Table 1*, along with typical ranges for their values used in the simulations. The relevant ratios of parameters are consistent with those observed experimentally (*Kim, 1996*; *Ma et al., 2005*).

## Additional information

### Funding

| Funder | Grant reference number | Author |
|---|---|---|
| Defense Advanced Research Projects Agency | Contract No. HR0011-16-C0062 | Dervis Vural |

The funders had no role in study design, data collection and interpretation, or the decision to submit the work for publication.

### Author contributions

Gurdip Uppal, Conceptualization, Data curation, Software, Formal analysis, Validation, Investigation, Visualization, Methodology, Writing—original draft, Writing—review and editing; Dervis Can Vural, Conceptualization, Resources, Supervision, Funding acquisition, Investigation, Visualization, Methodology, Writing—original draft, Project administration, Writing—review and editing

### Author ORCIDs

Gurdip Uppal (iD) http://orcid.org/0000-0003-3957-256X
Dervis Can Vural (iD) https://orcid.org/0000-0002-0495-8086

### Decision letter and Author response

Decision letter https://doi.org/10.7554/eLife.34862.026
Author response https://doi.org/10.7554/eLife.34862.027

## Additional files

### Supplementary files

• Source code 1. Matlab code for discrete stochastic simulations. Main file is main_discrete.m. See README file for more information.
DOI: https://doi.org/10.7554/eLife.34862.019

• Source code 2. Matlab code for continuous simulations used in *Figure 3B*. Main file is main_continuous.m. See README file for more information.

DOI: https://doi.org/10.7554/eLife.34862.020

• Transparent reporting form
DOI: https://doi.org/10.7554/eLife.34862.021

## Data availability

All data generated or analysed during this study are included in the manuscript and supporting files

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

## Appendix 1

DOI: https://doi.org/10.7554/eLife.34862.022

Here we describe a number of simplifying assumptions and following mathematical analysis that allow us to make sense of our simulation results, both qualitatively and quantitatively. Our approach is the standard Turing analysis commonly used to describe pattern formation in reaction diffusion systems. We first obtain a steady state solution, that is find out the population density and concentration of the public good and waste compound that leads to an equilibrium state. We then linearize the system around this equilibrium, and find out what kind of perturbations destabilize this equilibrium. The spotty patterns that form upon destabilization biologically correspond to cooperating microbial communities, the size of which we obtain in terms of system parameters. Our analytical outcomes are compared to discrete simulations in *Figure 3*.

Our simulations start with the whole population having a given secretion rate. The Turing analysis conducted below is then to find the group size and reproduction rate with this secretion rate.

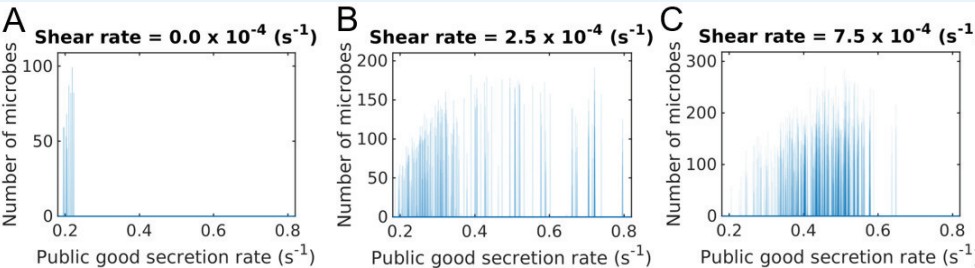

**Appendix 1—figure 1.** Secretion rate distributions of groups under different shear rates. Microbes are grouped by their position in space and the distribution of their secretion rates is plotted. Simulations were started with a secretion rate of 100 and random mutations were allowed to change the secretion rate of individual microbes. Distributions are given by looking at 10 simulations for each shear rate after a duration of $2.0 \times 10^5$ s. (**A**) With no shear, there are very few groups left at the chosen time and these will eventually go extinct. Cheaters have taken over these groups and individual groups are homogeneous in secretion rate. (**B**) With some added shear, groups are better able to beat cheating mutations, but the bulk of the distribution is still at lower secretion rates. (**C**) At higher shear rates there are more groups centered at a higher secretion rate due to shear augmented group fragmentation. In all cases, groups that initially start off with a heterogeneous population quickly become homogeneous, as seen by the delta peaks in the distributions. Each peak corresponds to a group. In the larger population however, groups of different secretion rates can and do coexist, as seen by the distribution of the peaks. In this case we can still analyze individual groups by looking at their secretion rate. The Matlab code and data for this figure is provided in *Appendix 1—figure 1—source data 1*.

DOI: https://doi.org/10.7554/eLife.34862.023

The following source data is available for figure :
**Appendix 1—figure 1—source data 1.** Matlab data and code files for figure in appendix.

DOI: https://doi.org/10.7554/eLife.34862.024

We also observe in our simulations, that groups are typically homogeneous in terms of secretion rate; that is once a cheating mutation occurs in a group, it fixates and quickly takes over the group, on a time scale much quicker than the group fragmentation (*Appendix 1—figure 1*). In other words, while the populations in different groups may have different secretion rates, the secretion rate within one group is approximately uniform. We can therefore simplify our equations to determine the spatial structure of a group in terms of the secretion rate of its constituents, even after mutations occur. To determine the native group

size and reproduction rate (i.e. when there is no shear and mutation) we can now write down a simplified set of equations,.

$$\frac{\partial n}{\partial t} = d_b \nabla^2 n + n \left[ \alpha_1 \frac{c_1}{c_1 + k_1} - \alpha_2 \frac{c_2}{c_2 + k_2} - \beta_1 s_1 \right] \tag{6}$$

$$\frac{\partial c_1}{\partial t} = d_1 \nabla^2 c_1 + n s_1 - \lambda_1 c_1, \tag{7}$$

$$\frac{\partial c_2}{\partial t} = d_2 \nabla^2 c_2 + n s_2 - \lambda_2 c_2. \tag{8}$$

Through some rescaling, we can non-dimensionalize the system. If we define the rescaled variables as

$$\begin{aligned}
&\vec{x} \leftarrow \vec{x} \sqrt{\frac{\lambda_1}{d_b}}, \quad t \leftarrow \lambda_1 t, \quad c_\alpha \leftarrow \frac{c_\alpha}{k_\alpha}, \quad d_\alpha \leftarrow \frac{d_\alpha}{d_b} \\
&s_\alpha \leftarrow \frac{s_\alpha}{k_\alpha \lambda_\alpha}, \quad \mu \leftarrow (k_1 \lambda_1)^2 \mu, \quad \alpha_1 \leftarrow \frac{\alpha_1}{\lambda_1}, \quad \alpha_2 \leftarrow \frac{\alpha_2}{\lambda_1}, \\
&\beta_1 \leftarrow \beta_1 k_1, \quad \vec{v} \leftarrow \vec{v} \sqrt{\frac{1}{\lambda_1 d_b}}, \quad \sigma \leftarrow \frac{\lambda_2}{\lambda_1},
\end{aligned} \tag{9}$$

then our equations become,

$$\frac{\partial n}{\partial t} = \nabla^2 n + n \left[ \alpha_1 \frac{c_1}{c_1 + 1} - \alpha_2 \frac{c_2}{c_2 + 1} - \beta_1 s_1 \right], \tag{10}$$

$$\frac{\partial c_1}{\partial t} = d_1 \nabla^2 c_1 + n s_1 - c_1, \tag{11}$$

$$\frac{\partial c_2}{\partial t} = d_2 \nabla^2 c_2 + n \sigma s_2 - \sigma c_2. \tag{12}$$

We will now obtain steady states and conditions for linear stability. We first obtain the steady states in the absence of diffusion and investigate the stability of the system. The steady states are given by

$$n \left[ \alpha_1 \frac{c_1}{c_1 + 1} - \alpha_2 \frac{c_2}{c_2 + 1} - \beta_1 s_1 \right] = n s_1 - c_1 = n \sigma s_2 - \sigma c_2 = 0.$$

This gives, either the trivial solution $n^\star = c_1^\star = c_2^\star = 0$, or the solutions:

$$c_\alpha^\star = n^\star s_\alpha, \quad n^\star = \frac{-b \pm \sqrt{b^2 - 4ac}}{2a},$$

where $a = (\alpha_1 - \alpha_2 - \beta_1 s_1) s_1 s_2$, $b = \alpha_1 s_1 - \alpha_2 s_2 - \beta_1 s_1 (s_1 + s_2)$, and $c = -\beta_1 s_1$. For this to be a sensible solution, we require $n^\star$ to be real and positive. This also imposes conditions on the system parameters.

Next, we establish the local stability of this solution by perturbing the system away from the steady state and expand up to first order. We take the perturbation $\mathbf{w} = (n, c_1, c_2)^T - (n^\star, c_1^\star, c_2^\star)^T$, and substitute it into our system to get the linear system

$$\frac{\partial}{\partial t} \mathbf{w} = A \mathbf{w}. \tag{13}$$

With the stability matrix, $A$, given as,

$$A = \begin{pmatrix} 0 & f_1 & f_2 \\ s_1 & -1 & 0 \\ \sigma s_2 & 0 & -\sigma \end{pmatrix},$$ (14)

where

$$f_1 = \frac{\alpha_1 n^\star}{(n^\star s_1 + 1)^2} \quad f_2 = -\frac{\alpha_2 n^\star}{(n^\star s_2 + 1)^2}.$$

The system is stable if the eigenvalues, $\Lambda$ of this matrix have a negative real part. The characteristic polynomial for the eigenvalues is given as $\Lambda^3 + A_0\Lambda^2 + B_0\Lambda + C_0 = 0$, where

$$A_0 = 1 + \sigma,$$
$$B_0 = \sigma - f_1 s_1 - f_2 s_2 \sigma,$$
$$C_0 = -\sigma(f_1 s_1 + f_2 s_2).$$

Since the first two coefficients of the characteristic equation are positive, by Descartes' rule of signs, in order to get only negative real part eigenvalues, we need $B_0$ and $C_0$ to be positive as well. This is a requirement for linear stability. Thus, we have the conditions

$$\sigma - f_1 s_1 - f_2 s_2 \sigma \geq 0,$$
$$-f_1 s_1 - f_2 s_2 \geq 0 \Rightarrow |f_1 s_1| \leq |f_2 s_2|.$$

Next we include diffusion and analyze the instability caused by diffusion. Fourier expanding the solution

$$\mathbf{W}(\mathbf{x}, t) = \sum_k c_k e^{i\mathbf{k}\cdot\mathbf{x}} e^{\Lambda(\mathbf{k})t},$$ (15)

and plugging this into our equations, we get the eigenvalue equation, $(-k^2 D + A)\mathbf{W} = \Lambda\mathbf{W}$, where

$$D = \begin{pmatrix} 1 & 0 & 0 \\ 0 & d_1 & 0 \\ 0 & 0 & d_2 \end{pmatrix}.$$ (16)

Solving for the eigenvalues again gives a characteristic equation of the form $\Lambda^3 + A\Lambda^2 + B\Lambda + C = 0$, where now

$$A = A_0 + \Theta(k^2),$$ (17)

$$B = B_0 + \Phi(k^2),$$ (18)

$$C = C_0 + \Psi(k^2),$$ (19)

and the $k^2$ dependent functions are,

$$\Theta(k^2) = [1 + d_1 + d_2]k^2,$$ (20)

$$\Phi(k^2) = [d_1 d_2 + d_1 + d_2]k^4 + [d_1\sigma + d_2 + \sigma + 1]k^2,$$ (21)

$$\Psi(k^2) = d_1 d_2 k^6 + [d_1\sigma + d_2]k^4 + [\sigma - d_2 f_1 s_1 - d_1 f_2 s_2 \sigma]k^2.$$ (22)

We can again use Descartes' rule of signs, this time looking for an instability, which will happen when $\Psi(k^2)$ is sufficiently negative. To be precise, the range of unstable wavenumbers satisfy,

$$\Psi(k^2) < -C_0. \tag{23}$$

At the critical values for the diffusion parameters, the function $\Psi(k^2) + C_0$ only vanishes at one point, the local minumum of $\Psi(k^2)$. This occurs at the critical $k^2$ value where $\mathrm{d}\Psi/\mathrm{d}k^2 = 0$, giving

$$k_{\mathrm{crit}}^2 = \frac{-b_k + \sqrt{b_k^2 - 4a_k c_k}}{2a_k}, \tag{24}$$

where $a_k = 3d_1 d_2, b_k = 2[d_1\sigma + d_2]$, and $c_k = \sigma - d_2 f_1 s_1 - d_1 f_2 s_2 \sigma$. We take the positive root in 33 corresponding to the physical, positive $k^2$.

For all combinations of parameters giving rise to stable groups, we observe in our simulations that the group size approximated well by $2\pi/k_{\mathrm{fast}}$, where, $k_{\mathrm{fast}}$ is the wave number corresponding to the fastest growing mode (i.e. the value that maximizes $\Lambda(k^2)$). The size of the microbial clusters, as obtained by analytical theory $2\pi/k_{\mathrm{fast}}$ and stochastic simulations are shown in the top row of 3. The color indicates the size of microbialgroups.

Roughly speaking, if we view a microbial cluster as the cause of perturbation at a nearby location, the Turing instability will manifest as group reproduction. We should strongly caution that as the instability proceeds, the system moves away from the initial unstable fixed point around which it was linearized, and thus the exponential dependence in 24 should eventually break down. Nevertheless, the eigenvalues in the exponents still provide us with an approximate estimation of the group reproduction rate, within a factor of two near the phase boundary.

