## [Decision Letter]

Thank you for submitting your work entitled "Shearing in flow environment promotes evolution of social behavior in microbial populations" for consideration by *eLife*. Your article has been evaluated by a Senior Editor and three reviewers, one of whom, Jeff Gore (Reviewer #3), served as Reviewing Editor. Our decision has been reached after consultation between the reviewers.

The referees appreciated that the study allows for a principled way of thinking about how spatial structure could arise. However, there were several significant concerns that were expressed regarding the presentation, conceptual framing, and technical implementation of the model. We do believe that the approach has potential, so if you can address all the concerns of the referees, we would consider a new manuscript.

Major issues that would have to be addressed:

* Framing of the question in terms of spatial structure vs. group selection

* Clarify how heterogeneity is analyzed in the population

* Does the social strain actually evolve / spread in the model?

* Show what happens with fixed strategies with and without flow and with and without the Turing pattern and then explore the effect of evolution

* Was there an error in the implementation of the model? (See reviewer #2's comments.)

*Reviewer #1:*

The manuscripts considers the problem of how cooperation survives in the presence of cheaters. The explanation is based on the Simpson's paradox (paper by Stan Leibler is not cited and the mechanism is not discussed). The novelty is that the metacommunity dynamics emerge from the governing equations rather than being externally imposed.

Specifically, the production of public good and waste are set up to produce a Turing instability, which creates patches of high population density separated by voids. These act as local communities in a meta community. Within each patch cooperators are outcompeted, but patches with many cooperators grow faster than patches with few cooperators. As a result, cooperators persist.

The key factor in the above argument is that the competition occurs between different patches. It seems to me that, for this to happen, the patches need to be constantly reformed. Reforming could occur due to breaking of the Turing patters by the flow. I think this mechanism is very important, but it is not discussed in the manuscript in any detail. Instead, a lot of the Discussion is focused on the enhancement of the cluster growth rate due to flow. While this enhancement plays a role, it is not clear whether this or the above mechanism is the dominant force. This seems to be a major point to address.

Other comments. The first half of the paper restates the standard results from Turing pattern formation and is better suited for Supplementary Information. I don't think these derivations are necessary to understand the main results and will probably be skipped by most readers. The simulation methods need to be better discussed. The following paper could also be relevant, Drescher, Knut, et al., 2014.

Overall, this seems to be a solid research paper. As far as potential impact, there are three distracting factors:

1) Simpson's paradox is pretty well established.

2) Turing patterns are known to occur in ecological models and possibly real ecosystems. The fact that inhomogeneous spatial densities due to Turing patterns affect species interactions is also well known. There is a lot of literature on this. See for example Wilson, Morris and Bronstein, 2003.

3) It is not clear whether the specific model is applicable to any specific ecosystem or that the Turing mechanism is robust and present in actual microbial communities.

*Reviewer #2:*

In this paper, the authors show that advection in the spatial population dynamics of microbes can lead to the assortment that is necessary to maintain cooperation (in the form of public good production).

This is potentially interesting, but I find the background and Introduction of the paper conceptually flawed. Moreover, the basic model seems to contain an error.

For me, the conceptual problems start with the first sentence of the Abstract: "It is advantageous for microbes to form social aggregates when they commonly benefit from secreting a public good." This is the wrong premise: whether it is advantageous to form groups is a question, not a fact.

Similarly, in the Introduction the authors state: "… until the lack of public goods compromise the fitness of the entire group". This sounds like it is already known how this plays out, but that's exactly the problem: it is generally *not* known how this plays out. The way the authors are stating this, the outcome would be a matter of the relative strength of individual vs. group selection.

However, despite the fact that they are vaguely referring to "group fitness" and "species fitness" throughout the paper, their model does not have reproducing groups, and hence they are not talking about group selection (see also below). In fact, they are talking about individual selection, and in particular they are talking about a particular mechanisms, advection, that can lead to assortment of cooperative types and thereby the maintenance of cooperation.

In general, the Introduction doesn't make much conceptual sense to me. For example, I don't see how quorum sensing falls into the category of mechanism where altruism is a consequence of a self-serving trait. With quorum sensing, the problem is shifted onto the production of the sensing signal, which becomes itself a public good. Judging from the literature cited, the authors do not seem to be up to date with group selection theory: Simon et al., 2012, 2013 have shown that kin selection is *not* closely linked to kin selection, as claimed by the authors in the third paragraph of the Introduction.

Group selection is an altogether different mechanism than selection at the individual level, of which kin selection is an example. In general, all individual-level explanations for cooperation, including kin selection, can be understood on the basis of assortment (Fletcher and Doebeli 2009). Group selection is a conceptually different mechanism that involves differential birth and death of groups with different type compositions, as well as (possibly) interactions between groups (Simon et al., 2013). In particular, it seems clear that the mechanism for the evolution of cooperation proposed by the authors is an example of (spatial) assortment, and not of group selection as they claim. They mention that in their model, spatial groups form and "reproduce", but this is simply an emergent property of the model, and the significance of the differential reproductive success of different groups is not investigated in the paper. My guess is that even if these "groups" would not reproduce, cooperation would still be maintained due to the assortment caused by advection. Overall, it is therefore not clear that the authors fully understand the conceptual underpinnings of the evolution of cooperation, and how their model fits with existing theory and concepts.

Perhaps more importantly, I think the basic model 1-3 may contain a fundamental flaw. Specifically, in Equation 2 the term -*ns_1_* does not seem to make sense, since this represents only the public good produced by those microbes whose cooperation level is *s_1_*. This term should be replaced by an integral over *s_1_*, thus measuring the contribution to the public good of all types. Similarly, the term -*ns_2_* in Equation 3 seems wrong, as here the term *n* should be replaced by an integral over all types *s_1_*, i.e., n should be replaced by the total number of all microbes living at a given location. (Note that the model does not assume only a single type *s_1_*at any given location; otherwise the second derivative with respect to *s_1_*in Equation 1, i.e., diffusion in *s_1_*space, would not make sense.) Given that the basic equations may be wrong, I am uneasy about the rest of the analysis in the paper.

*Reviewer #3:*

Overall, I found this to be an intriguing combination of two rather different fields. As the authors describe, there is a long history of theory and experiment probing the conditions required for the evolution of cooperative behaviors, and some of this work has focused on spatial structure. However, little of the work has considered interesting flow fields with shear. I very much like the idea of a public good and a public bad, each of which is secreted. Then, if the public bad diffuses faster than the public good it is possible to have Turing instabilities.

My primary concern is that I didn't understand how the dynamics between cooperation and cheating took place in terms of distribution and abundance of the different strategies. How much cooperation was present in the various flow situations? Was there heterogeneity in the population? What would happen if there were a (non-evolving) fixed cooperator and another genetic cheater?

[Editors’ note: what now follows is the decision letter after the authors submitted for further consideration.]

Thank you for resubmitting your work entitled "Shearing in flow environment promotes evolution of social behavior in microbial populations" for further consideration at *eLife*. Your revised article has been evaluated by Arup Chakraborty (Senior Editor), and three reviewers, one of whom served as Reviewing Editor

The manuscript has been improved but there are remaining issues that need to be addressed before the manuscript can be considered for acceptance, as outlined below:

Two of the three reviewers are still uncomfortable with your treatment of heterogeneity in the population. Even if a cheater will quickly take over a local population, it does not mean that there cannot be heterogeneity of different levels of cooperation within each local population or within the broader metapopulation, where different groups could have different levels of cooperation. As far as we can tell, this sort of heterogeneity was never analyzed in the paper, but we think that it is very important for a complete understanding of the system. As indicated below in more detail, this is not simply a question of "notation." It seems that the integral was added into the model description but that no analysis of this heterogeneity was actually done.

Please consider the following points in revising your manuscript:

1] The authors state that "It is well known that evolution of altruism in species strongly depends on the individuals being discrete (Durrett and Levin (1994)). This would seem to imply that deterministic models, e.g. involving differential equations for frequencies of different types, would never lead to cooperation, which is of course not true. Furthermore, the argument in Simon et al. (2012) for why group selection is different from kin selection is *not* primarily based on the asynchrony of individual-level and group-level events. Rather, it is a consequence of the fact that assortment (or, equivalently, relatedness) may have no influence on certain group level events, such as games between groups.

2] The authors' statement in the rebuttal letter that "This change in notation does not affect our results or analysis, since in our simulations we start the population with a fixed secretion rate." is unclear to us. First, starting with a fixed secretion rate would not make the original equations correct, because mutation would quickly lead to a situation in which secretion rates are variable. Second, what is meant by "a change in notation does not affect our results"? Of course, a change in notation does not affect anything, apart from the notation in itself. The question is whether the simulations were carried out correctly, i.e., with the integral (summation) term taken into account. Have the simulations been carried out according to the new, correct equations?

3] The authors' finding that without advection, cooperation cannot be maintained in their system is curious. This seems contrary to previous work by Hauert and colleagues (Wakano et al., 2009), which shows that cooperation is maintained in public goods reaction-diffusion systems. I am wondering about the cause of the different outcomes in these models. Is it due to the continuous nature of the secretion trait considered in the present study? If so, this might be important to point out in the Discussion.

---

## [Author Response]

[Editors’ note: the author responses to the first round of peer review follow.]

Reviewer #1:

The manuscripts considers the problem of how cooperation survives in the presence of cheaters. The explanation is based on the Simpson's paradox (paper by Stan Leibler is not cited and the mechanism is not discussed). The novelty is that the metacommunity dynamics emerge from the governing equations rather than being externally imposed.

We have now added a reference to Stan Leibler’s paper [1] in the Introduction and final discussion citing that our results also fall under the broad phenomenon of Simpson’s paradox (Introduction, eleventh paragraph, Discussion, second paragraph).

Specifically, the production of public good and waste are set up to produce a Turing instability, which creates patches of high population density separated by voids. These act as local communities in a meta community. Within each patch cooperators are outcompeted, but patches with many cooperators grow faster than patches with few cooperators. As a result, cooperators persist.The key factor in the above argument is that the competition occurs between different patches. It seems to me that, for this to happen, the patches need to be constantly reformed. Reforming could occur due to breaking of the Turing patters by the flow. I think this mechanism is very important, but it is not discussed in the manuscript in any detail. Instead, a lot of the Discussion is focused on the enhancement of the cluster growth rate due to flow. While this enhancement plays a role, it is not clear whether this or the above mechanism is the dominant force. This seems to be a major point to address.

We have now added more discussion to clarify that it is indeed the shear driven group fragmentation that promotes sociality, and not the growth rate of the clusters (subsection “Social groups as Turing patterns”). We have also modified Figure 2 to better illustrate the main mechanism, and have included videos of our simulations in Supplementary files 1-5. Also, Figure 4A shows that group fragmentation rate increases with shear.

Other comments. The first half of the paper restates the standard results from Turing pattern formation and is better suited for Supplementary Information. I don't think these derivations are necessary to understand the main results and will probably be skipped by most readers. The simulation methods need to be better discussed. The following paper could also be relevant, Drescher, Knut, et al., 2014.

We have now moved the mathematical details to an appendix to help clarify the biological results. We have added more detail in the Materials and methods section (fourth paragraph) explaining our simulation methods. We have also added a reference to Drescher’s paper [2] (Introduction, seventh paragraph).

Overall, this seems to be a solid research paper. As far as potential impact, there are three distracting factors:1) Simpson's paradox is pretty well established.2) Turing patterns are known to occur in ecological models and possibly real ecosystems. The fact that inhomogeneous spatial densities due to Turing patterns affect species interactions is also well known. There is a lot of literature on this. See for example Wilson, Morris and Bronstein, 2003.3) It is not clear whether the specific model is applicable to any specific ecosystem or that the Turing mechanism is robust and present in actual microbial communities.

Regarding the “distracting factors” mentioned above, we agree that Simpson’s paradox and Turing patterns in nature are well known. We now emphasize in the paper, that the novelty is not in the Turing mechanism. This is our starting point (Introduction, eleventh paragraph). We now clarify in the paper that the novel aspect of our work is the effect of fluid shear on spatial assortment and on the evolution of cooperation (Introduction, twelfth paragraph). We’ve also added reference to Wilson’s paper [3] (Introduction, eleventh paragraph and subsection “Social groups as Turing patterns”, first paragraph) as well as references to studies finding growth patterns in microbial systems [4,5].

As for point number three: We would be very excited to see experimental verification of our fluid dynamical model in specific aquatic niches. We now emphasize that we have built our model and chosen parameter values (e.g. diffusion constants, decay rates, flow velocities etc.) from well-established results (Materials and methods, third and last paragraphs) that would generalize to a wide range of systems (Introduction, tenth paragraph). Hopefully this generality will inspire experimentalists to look for the phenomenon we reported.

1] Chuang, John S., Olivier Rivoire, and Stanislas Leibler. "Simpson's paradox in a synthetic microbial system." Science323.5911 (2009): 272-275.

2] Drescher, Knut, et al. "Solutions to the public goods dilemma in bacterial biofilms." Current Biology 24.1 (2014): 50-55.

3] Wilson, W. G., W. F. Morris, and J. L. Bronstein. "Coexistence of mutualists and exploiters on spatial landscapes." Ecological monographs 73.3 (2003): 397-413.

4] Ben-Jacob, Eshel, et al. "Generic modelling of cooperative growth patterns in bacterial colonies." Nature 368.6466 (1994): 46-49.

5] Chang-Li, Xie, et al. "Microcalorimetric study of bacterial growth." Thermochimica Acta 123 (1988): 33-41.

Reviewer #2:

In this paper, the authors show that advection in the spatial population dynamics of microbes can lead to the assortment that is necessary to maintain cooperation (in the form of public good production).This is potentially interesting, but I find the background and Introduction of the paper conceptually flawed. Moreover, the basic model seems to contain an error.For me, the conceptual problems start with the first sentence of the Abstract: "It is advantageous for microbes to form social aggregates when they commonly benefit from secreting a public good." This is the wrong premise: whether it is advantageous to form groups is a question, not a fact.Similarly, in the Introduction the authors state: "… until the lack of public goods compromise the fitness of the entire group". This sounds like it is already known how this plays out, but that's exactly the problem: it is generally not known how this plays out. The way the authors are stating this, the outcome would be a matter of the relative strength of individual vs. group selection.

We have now changed the Abstract to state that the general question is how microbes maintain cooperative behavior. Additionally, we state (subsection “Social groups as Turing patterns”) that one way to maintain this behavior is through forming social groups. In our model, without this spatial structuring, the population does get taken over by cheaters. We have also modified Figure 2 to illustrate this phenomenon in a clearer way. We have also included videos of simulations in the different scenarios in Supplementary files 1-5.

However, despite the fact that they are vaguely referring to "group fitness" and "species fitness" throughout the paper, their model does not have reproducing groups, and hence they are not talking about group selection (see also below). In fact, they are talking about individual selection, and in particular they are talking about a particular mechanisms, advection, that can lead to assortment of cooperative types and thereby the maintenance of cooperation.

We clarify that we indeed have reproducing and dying groups in our system, these are emergent phenomena from the individual level dynamics, this is evident from (Equations 6, 9, 10) and also, Figure 4A gives the group reproduction rate as a function of shear. To clarify this, we have also added some more discussion (Introduction, eleventh paragraph and subsection “Social groups as Turing patterns”). Also, we have now modified Figure 2 to explicitly show how shear driven group fragmentation enhances cooperation. The videos included in the supplementary files also illustrate this. Hopefully this should greatly clarify our main finding.

In general, the Introduction doesn't make much conceptual sense to me. For example, I don't see how quorum sensing falls into the category of mechanism where altruism is a consequence of a self-serving trait. With quorum sensing, the problem is shifted onto the production of the sensing signal, which becomes itself a public good.

We have clarified this discussion to explain how quorum sensing can be an example of reciprocity (Introduction, third paragraph); the cited paper by Allen 2016, discusses this phenomenon.

Judging from the literature cited, the authors do not seem to be up to date with group selection theory: Simon et al., 2012, 2013 have shown that kin selection is not closely linked to kin selection, as claimed by the authors in the third paragraph of the Introduction.Group selection is an altogether different mechanism than selection at the individual level, of which kin selection is an example. In general, all individual-level explanations for cooperation, including kin selection, can be understood on the basis of assortment (Fletcher and Doebeli 2009). Group selection is a conceptually different mechanism that involves differential birth and death of groups with different type compositions, as well as (possibly) interactions between groups (Simon et al., 2013). In particular, it seems clear that the mechanism for the evolution of cooperation proposed by the authors is an example of (spatial) assortment, and not of group selection as they claim. They mention that in their model, spatial groups form and "reproduce", but this is simply an emergent property of the model, and the significance of the differential reproductive success of different groups is not investigated in the paper. My guess is that even if these "groups" would not reproduce, cooperation would still be maintained due to the assortment caused by advection. Overall, it is therefore not clear that the authors fully understand the conceptual underpinnings of the evolution of cooperation, and how their model fits with existing theory and concepts.

The review by Kramer [1] shows that this is still not a settled discussion. The model proposed by Simon [2, 3] makes an interesting point that since individual and group level selection are asynchronous in nature, they cannot be equivalent. We agree that according to the definition given by [2], group selection is different from kin selection, though there is still some debate on this [1, 4-6]. Nevertheless, we agree that our mechanism can be better described by individual level selection, and rather than muddle the discussion with group selection, we have changed our discussion throughout from group selection to individual selection and spatial assortment. Indeed, the fluid dynamical / diffusion equations on which the simulations are based on describes the growth and evolution of individual microbes, and not of groups. We therefore change our discussion to now focus on assortment that arises from population viscosity and budding dispersal, which, in our case, emerges from the Turing instability. We clarify this further with discussion (Introduction, fourth and fifth paragraphs) and have added references to Simon [2, 3] as well as Fletcher and Doebeli [7]. We also emphasize that cooperation would not be maintained without group fragmentation. Advection only enhances the group fragmentation, whereas the spatial isolation of microbial groups caused by the Turing instability is what allows for cooperation to persist (subsection “Social groups as Turing patterns”). We have now modified Figure 2 to illustrate this.

Perhaps more importantly, I think the basic model 1-3 may contain a fundamental flaw. Specifically, in Equation 2 the term -ns_1_ does not seem to make sense, since this represents only the public good produced by those microbes whose cooperation level is s_1_. This term should be replaced by an integral over s_1_, thus measuring the contribution to the public good of all types. Similarly, the term -ns_2_ in Equation 3 seems wrong, as here the term n should be replaced by an integral over all types s_1_, i.e., n should be replaced by the total number of all microbes living at a given location. (Note that the model does not assume only a single type s_1_ at any given location; otherwise the second derivative with respect to s_1_ in Equation 1, i.e., diffusion in s_1_ space, would not make sense.) Given that the basic equations may be wrong, I am uneasy about the rest of the analysis in the paper.

The referee is correct. The old version of the equation did not notate the continuum of possible secretion rates. We have now changed the equations of the model in the Materials and methods section, where our notation now accounts for a continuum of possible secretion rates. This change in notation does not affect our results or analysis, since in our simulations we start the population with a fixed secretion rate. The Turing analysis conducted in the paper is to find the native group size and reproduction rate with this secretion rate. We also observe in our simulations that groups are typically homogeneous in terms of secretion rate; i.e. once a cheating mutation occurs in a group, it fixates and quickly takes over the group, on a time scale much quicker than the group fragmentation. Therefore, this simplification is also applicable after mutations occur.

To better clarify our analysis, we have now moved the mathematical analysis into the appendix. We have added discussion in the first section of the appendix on how we can make some simplifying assumptions to the model to analyze homogenous groups (Appendix 1, first to fourth paragraphs). We also better clarify our simulation methods and acknowledge that we do not expect perfect agreement between the stochastic-discrete simulations and the continuous equations, although we were able to obtain the relevant quantities such as group size and reproduction rate, to good approximation, using a continuum analysis (Materials and methods, fourth paragraph).

1] Kramer, Jos, and Joël Meunier. "Kin and multilevel selection in social evolution: a never-ending controversy?." F1000Research 5 (2016).

2] Simon, Burton, Jeffrey A. Fletcher, and Michael Doebeli. "Towards a general theory of group selection." Evolution 67.6 (2013): 1561-1572.

3] Simon, Burton, Jeffrey A. Fletcher, and Michael Doebeli. "Hamilton's rule in multi-level selection models." Journal of theoretical biology 299 (2012): 55-63.

4] West, Stuart A., Ashleigh S. Griffin, and Andy Gardner. "Social semantics: altruism, cooperation, mutualism, strong reciprocity and group selection." Journal of evolutionary biology 20.2 (2007): 415432.

5] Gardner, Andy. "The genetical theory of multilevel selection." Journal of evolutionary biology 28.2 (2015): 305-319.

6] Goodnight, C. J. "Multilevel selection theory and evidence: a critique of Gardner, 2015." Journal of evolutionary biology 28.9 (2015): 1734-1746.

Fletcher, Jeffrey A., and Michael Doebeli. "A simple and general explanation for the evolution of altruism." Proceedings of the Royal Society of London B: Biological Sciences276.1654 (2009): 13-19

Reviewer #3:

Overall, I found this to be an intriguing combination of two rather different fields. As the authors describe, there is a long history of theory and experiment probing the conditions required for the evolution of cooperative behaviors, and some of this work has focused on spatial structure. However, little of the work has considered interesting flow fields with shear. I very much like the idea of a public good and a public bad, each of which is secreted. Then, if the public bad diffuses faster than the public good it is possible to have Turing instabilities.My primary concern is that I didn't understand how the dynamics between cooperation and cheating took place in terms of distribution and abundance of the different strategies. How much cooperation was present in the various flow situations? Was there heterogeneity in the population? What would happen if there were a (non-evolving) fixed cooperator and another genetic cheater?

We thank the referee for his interest and feedback. We have added discussion to clarify how heterogeneity was analyzed, in both the appendix (lines 436-460), and in the results (subsection “Social groups as Turing patterns”). We explain how spatial homogeneity would lead to take over by cheaters, and how Turing patterns are a necessary first step to have social evolution in our model. We also state that, because of the spatial structure, the cheaters do not enter cooperative groups very easily and remain localized in their own groups. Therefore, if there is no mutation, and only a fixed cheater, groups with cheaters would get taken over and die out, leaving the groups with cooperators. These groups are stable. However, there are random mutations, sociality can only prevail if the groups fragment faster than new mutants materialize (subsection “Social groups as Turing patterns”). We have also modified Figure 2 and included videos in Supplementary files 1-5 to better illustrate this main result.

[Editors' note: the author responses to the re-review follow.]

[…] Please consider the following points in revising your manuscript:1] The authors state that "It is well known that evolution of altruism in species strongly depends on the individuals being discrete (Durrett and Levin (1994)). This would seem to imply that deterministic models, e.g. involving differential equations for frequencies of different types, would never lead to cooperation, which is of course not true.

Deterministic models can and do lead to cooperation. What was shown by Durrett and Levin is that discreteness can alter the outcome. For example, in our particular case, having a continuous population density can allow for the existence of “micro-mutant populations”’ which can spread easier between adjacent groups. Furthermore, the discreteness fully separates the clusters of microbes from each other, since there cannot exist fractional individuals. In reality, microbes are discrete, and we thus expect a discrete simulation to better model the biology. We have clarified this discussion in the paper (subsection “Social groups as Turing patterns”, last paragraph).

Furthermore, the argument in Simon et al. (2012) for why group selection is different from kin selection is not primarily based on the asynchrony of individual-level and group-level events. Rather, it is a consequence of the fact that assortment (or, equivalently, relatedness) may have no influence on certain group level events, such as games between groups.

We agree that this additional distinction also separates group selection events from kin selection ones. We have further clarified this in the paper (Introduction, fifth paragraph).

2] The authors' statement in the rebuttal letter that "This change in notation does not affect our results or analysis, since in our simulations we start the population with a fixed secretion rate." is unclear to us. First, starting with a fixed secretion rate would not make the original equations correct, because mutation would quickly lead to a situation in which secretion rates are variable. Second, what is meant by "a change in notation does not affect our results"? Of course, a change in notation does not affect anything, apart from the notation in itself. The question is whether the simulations were carried out correctly, i.e., with the integral (summation) term taken into account. Have the simulations been carried out according to the new, correct equations?

In the first version of our paper there was a typo where integrals (summations) in Equations 2 and 3 were missing. However, simulations did take into account the sum. To be entirely clear, and to fully reassure the referees, we outline what the simulations do: Every microbe secretes a waste compound and a public good, and there are individual variations in the secretion rates of the latter (but not the former). To determine the spatial concentrations of these two compounds, we use the diffusion equation, treating each microbe as a source. The contribution of every individual, taking into account the individual variations in their secretion rates, is summed over.

Since diversity within and across groups is an interesting subject we have now added some analysis of the heterogeneity of secretion rates. We have prepared Figure 5 and extra panels to Figure 6 to show the average secretion rate of the population in the different flow situations. Appendix 1: Figure 1 also illustrates the distribution of secretion rates in the system. Every peak corresponds to one group. We observe that while there is a wide diversity in cooperative behavior *across* groups, there is very little diversity *within* a group. This is because among two phenotypes the less cooperative one will always dominate the more cooperative one, prohibiting stable coexistence of phenotypes within a group.

A final remark: There is still not a sum in appendix Equations 7, 8. This is because our analytical work has been significantly more difficult than simulations, and we were able to obtain formulas only when diversity in social behavior is ignored (and hence the reason for the original typo).

3] The authors' finding that without advection, cooperation cannot be maintained in their system is curious. This seems contrary to previous work by Hauert and colleagues (Wakano et al., 2009), which shows that cooperation is maintained in public goods reaction-diffusion systems.

We suspect the difference between our model and Hauert et al. might be the mutation rate. We find that flow shear is absolutely necessary for stable cooperation to exist beyond a critical mutation rate. We now added a calculation of this critical mutation rate and compared it with our simulations (the agreement is within 25%) (–subsection “Evolution of sociality in constant shear for a binary phenotype”, last paragraph).

I am wondering about the cause of the different outcomes in these models. Is it due to the continuous nature of the secretion trait considered in the present study? If so, this might be important to point out in the Discussion.

Our findings are not sensitive to the discrete vs. continuous social phenotypes. To show this clearly, we now run two sets of simulations: In one set, the individuals are allowed to have a continuum of social behaviors, whereas in the other set they are binary (either cooperative or cheating). We have now included results in our paper that supports our findings for both models (Figure 4, 5).